# Exploring Recommendations for Child and Adolescent Fundamental Movement Skills Development: A Narrative Review

**DOI:** 10.3390/ijerph20043278

**Published:** 2023-02-13

**Authors:** Wesley O’Brien, Zeinab Khodaverdi, Lisa Bolger, Orla Murphy, Conor Philpott, Philip E. Kearney

**Affiliations:** 1Sports Studies and Physical Education Programme, School of Education, University College Cork, 2 Lucan Place, Western Road, T12 K8AF Cork, Ireland; 2Department of Biobehavioral Studies, Teachers College, Columbia University, New York, NY 10027, USA; 3Department of Sport, Leisure and Childhood Studies, Munster Technological University (Cork Campus), T12 P928 Cork, Ireland; 4Sport & Human Performance Research Centre, Department of Physical Education and Sport Sciences, University of Limerick, V94 T9PX Limerick, Ireland

**Keywords:** motor skills, school, physical education, teaching, coaching, youth

## Abstract

Fundamental movement skills (FMS) are frequently referred to as the *“building blocks”* of movement for children and adolescents in their lifelong physical activity journey. It is critical, however, that FMS are developed within Physical Education learning environments and other sport-related settings, specifically as these building blocks of movement require appropriate teaching and practice opportunities. While FMS are well-established as an *“important focus”* for children and adolescents, to the authors’ knowledge, there appears to be no standardized FMS development guidelines existent within the literature. This paper will examine whether the frequency, intensity, time, and type (FITT) principle could be transferable to interventions focusing on FMS development, and if so, whether sufficient consistency of findings exists to guide practitioners in their session design. Applying the FITT principle in this way may help to facilitate the comparison of FMS-related intervention studies, which may contribute to the future development of practical FMS-related guidelines for children and adolescents.

## 1. Introduction

Fundamental movement skills (FMS) are common motor activities with a general goal [1], which involve the use of large muscle groups within the body [2,3]. In terms of their categorical distinction, FMS are often differentiated into three subsets: (1) locomotor skills (e.g., running and skipping); object-control/ball skills (e.g., throwing and catching); and (3) stability skills (e.g., balancing and twisting) [2]. As an important component of the motor-development domain, FMS facilitate participation in physical activity and contribute to the holistic development of children and adolescents [1,2,3]. Previous evidence suggests that the development of FMS proficiency in children and adolescents can serve as the foundational building blocks for future lifelong daily activities [4]. FMS proficiency increases the likelihood of children and adolescents developing specialized movement sequences for participating effectively across a variety of organized and unorganized physical activities [2].

FMS are commonly viewed as a central tenet and developmental stage within the motor-skills domain [2,5]. In terms of Physical Education (PE), physical activity (PA), and sport settings, FMS have a critical role in both promoting and maintaining healthy developmental trajectories in children and adolescents [6]. In terms of empirical health-related research associations, positive relationships have been consistently found between FMS and PA, physical fitness, body composition, self-belief, and executive functioning [7,8].

While these positive associations between FMS and health have been observed quite frequently within the literature, children and adolescents do not solely acquire these motor skills as a result of the maturation process [9,10]. The development of FMS proficiency in children and adolescents is often dependent on the quality of the instructional environment and the provision of practice-based opportunities [2,11], augmenting the importance of key stakeholders, such as PE teachers, sport pedagogues, coaches, and researchers, within this process.

Research and practice have consistently highlighted how FMS interventions are successful in promoting skills and physical health [12,13,14]; however, the specific parameters for FMS-related recommendations have not been systematically investigated in PE, PA, and sport settings. The evidence to date has identified various strategies for measuring FMS-training exposure (e.g., types of FMS interventions) in children and adolescents; however, the general concepts for quantifying the frequency, duration (time), and intensity of FMS-related training [15] is less known among PE teachers, coaches, and sport practitioners. Together, the combined frequency, intensity, time, and type (FITT) product [16] has the potential to yield an evidence-informed FMS-related training dose for children and adolescents. The well-established FITT principle has been traditionally used to characterise recommended guidelines for PA and exercise [17].

Despite the growing number of FMS-related interventions in PE, PA, and sport settings, there appears to be an inconsistency as to *“what works”* in terms of an appropriate FMS-related training dosage [13]. The development of FMS in children and adolescents was empirically and correctly established as *“an important focus”* in 2016 [11]. Now, what remains for PE teachers, coaches, sport pedagogues, and researchers is the implementation of evidence-informed FMS-related recommendations. The FITT training principle may be one such way of providing this implementation guidance for practitioners, particularly once sufficient evidence is identified across studies to develop appropriate recommendations. Indeed, other health-and-exercise-related fields have previously used the FITT principle to categorise successful intervention features and provide subsequent recommendations for practitioners [18,19]. This paper, therefore, seeks to examine whether FMS interventions present sufficient consistency to be summarised and modified using the FITT principle, with the overall aim of providing evidence-informed and practical FMS-related recommendations for practitioners.

## 2. Methods Section

As part of this narrative review, a search was conducted using seven databases, including PubMed, MEDLINE, SPORTDiscus, CINAHL, Scopus, Web of Science, and EMBASE, without any date restriction for articles pertaining to motor skills interventions in typically developing children and adolescents from all socioeconomic backgrounds.

The main search group terms were: “fundamental motor skills” OR “FMS” OR “motor skill” OR “movement skill” OR “motor development” OR “motor performance” OR “balance” OR “stability” OR “motor ability” AND “children” OR “adolescent” OR “youth”, OR “preschooler” AND “intervention” OR “program” OR “study” OR “trial”. To exclude studies that specifically examined youth participants with disorders/disabilities, the following terms were used: AND NOT “disability” OR “disorder” OR “autism” OR “impairment” OR “cerebral palsy”.

The criteria used to include a study in this narrative were as follows: (1)The study needed to measure motor skill performance as an outcome and include pre- or post-timepoint intervention assessments, or both. It is important to note that neither the type of motor skill battery assessment tool nor the measurement approach was a determining factor in either including or excluding a study;(2)Only articles published in English and in peer-reviewed journals were considered. All books, reviews, theses, dissertations, commentaries, qualitative studies, and case studies were excluded from the review process as part of this narrative review.

Section 3 below provides information as part of this narrative review on the included FITT components of the motor skill intervention studies that have been reviewed.

## 3. Components of FITT Principle

### 3.1. Frequency

The “F” within the FITT principle stands for frequency and relates to how often a person participates in exercise-related training sessions [15]. To determine the frequency of FMS-related training sessions or interventions, a specific priority in this research was placed towards published randomized controlled trials (RCTs) in FMS, specifically as they are the most likely research designs to provide impartial information on the frequency variable [20]. As part of this specifically tailored narrative research search, a total of 36 RCTs were retrieved across a 25-year timeline (1997–2022), with 88% of the published research reporting on frequency-related FMS data. From the critical review of this RCT research, Table 1 documents the frequency of FMS-related training sessions and interventions, with the existing frequency evidence ranging from one to five FMS sessions a week. Interestingly, Table 1 further highlights that 8 of the existing 40 RCTs do not appear to report the frequency of their associated FMS-related training sessions or interventions.

Outside of this specific RCT search in FMS, other systematic review evidence has further attempted to synthesize the effectiveness of FMS-related interventions (RCTs and non-RCTs) on motor skill development [12,13,21,22,23]. Aligned to the data reported above in Table 1, the findings from Wick et al. (2017) similarly reported a frequency range of one to five FMS sessions a week for children and adolescents [13].

**Table 1 ijerph-20-03278-t001:** The reported frequencies of FMS-related training sessions/interventions from randomized controlled trial evidence from 1997 to 2022.

No. FMS Sessions/Week	1 × FMS Session/Week	2 × FMS Sessions/Week	3 × FMS Sessions/Week	4 × FMS Sessions/Week	5 × FMS Sessions/Week	Frequency Not Reported
	**6 studies** Cliff et al. (2011) [24] Foulkes et al. (2017) [25] Johnson et al. (2019) [26] McGrane et al. (2018) [27] Pesce et al. (2016) [28] Smyth & Q′Keeffe (1998) [29]	**16 studies** Derri et al. (2001) [30] Donath et al. (2015) [31] Gallotta et al. (2017) [32] Goodway & Branta, (2003) [33] Goodway et al. (2003) [34] Hamilton et al. (1999) [35] Iivonen et al. (2011) [36] Johnson et al. (2019) [26] Marshall & Bouffard, (1997) [37] Palmer et al. (2019) [38] Roach & Keats (2018) [39] Robinson & Goodway (2009) [40] Robinson et al. (2017) [41] Veldman et al. (2017) [42] Zask et al. (2012) [43] Berleze & Valentini (2022) [44]	**3 studies** Hashemi et al. (2015) [45] Jones et al. (2011) [46] Robinson et al. (2022) [47]	**2 studies** De Oliveira et al. (2019) [48] Hestbaek et al. (2021) [49]	**5 studies** Alhassan et al. (2012) [50] Engel et al. (2018) [51] Roth et al. (2015) [52] Webster et al. (2020) [53] Staiano et al. (2022) [54]	**8 studies** Chan et al. (2019) [55] Cohen et al. (2015) [56] Lander et al. (2017) [57] McKenzie et al. (2002) [58] Miller et al. (2015) [59] Salmon et al. (2008) [60] Trost & Brookes (2021) [61] van Beurden et al. (2003) [62]

Note: No. = number; sessions/week = sessions a week; FMS = fundamental movement (motor) skills.

Overall, the evidence presented above suggests that a frequency of 2 days per week for FMS-related activities appears to be somewhat prevalent and commonly reported in many studies [13,20]. Furthermore, researchers and practitioners ought to consider if there is an optimal FMS-related frequency focus per week, which might result in maximal skill acquisition outcomes in children and adolescents. At this point, it seems to be unclear as to whether the interaction of FMS-related frequencies with the other FITT components (intensity, time, and type) may impact the quality of FMS acquisition. While some studies report having a varying number of FMS-related training sessions or interventions per week, other research illustrates the total volume time (minutes) of FMS-related activities per week, irrespective of the frequency variable [61]. In this sense, therefore, the important debate of frequency versus time for FMS needs more clarification and guidance for those working with children and adolescents in a physically active setting.

### 3.2. Intensity

The “I” within the FITT principle stands for “intensity”. Within the context of PA, intensity refers to the energy expended during a given time period [63]. In the context of FMS session design, however, another interpretation of intensity might also be considered as the number of skill executions within a given time period. Children’s practice of FMS, whether in the context of a PE lesson or a coaching session, will ideally achieve the dual objectives of reaching moderate-to-vigorous levels of PA for immediate health benefits while also promoting quality skill development [62]. With appropriate activity design and monitoring of intensity, both of these objectives can be met within the same session.

For children and adolescents between the ages of 5 and 17 years old, the World Health Organisation (WHO) recommend 60 min of moderate-to-vigorous physical activity (MVPA) daily [64]. There are five categories of well-established PA-related exercise intensities, beginning with *sedentary*, which typically refers to sitting or other stationary activities, requiring minimal energy and low levels of movement [65]. *Light*-*intensity* activity refers to activities that can be easily sustained for 60 min without incurring a noticeable change in breathing rate, and often, these activities require less than three times the resting energy expenditure [65]. *Moderate intensity* refers to using between 3 and 6 times more energy than a resting state, while *vigorous intensity* refers to using 6 to 9 times more energy when compared to a resting state [65]. *High-intensity* activity, however, refers to expending energy in excess of 9 times the amount used at rest.

Monitoring PA intensity is a critical component for evaluating cardiorespiratory fitness, and consistent participation at higher-intensity PA has the potential to positively impact other health markers, such as blood glucose and blood lipid levels [66]. The consistent associations found between FMS and PA indicate that a high level of motor competence can contribute to long-term PA engagement [67]. Minimal research, however, exists pertaining to the intensity of activity accrued during the performance of FMS activities [68] or the associated quality of skill execution.

Recent studies with populations of children and adolescents have examined energy expenditure during the performance of object-control skills (kick, throw, strike) [68,69]. The Sacko et al. (2019) study (*n* = 42; 22 males; mean age = 8.1 ± 0.8 years) reported that the practice of kicking, throwing, and striking at a rate of two maximal-effort attempts per minute appears to meet the threshold for moderate-intensity PA as it surpasses the 4.0 metabolic equivalent (METs) [69]. Further research among children (*n* = 30; 16 males; (9.4 ± 1.4 years) has suggested a slow cadence (i.e., kicking a football in a passing motion every 6 s but not at maximal effort), results in a light intensity of activity between 1.5 and 2.9 METs [70]. Notably, these studies differ in the effort applied from participants; the Sacko et al. (2019) study sought maximal effort from participants, whereas the Duncan et al. (2020) study referred to a short passing motion with a football at a slow tempo [69,71]. Such tempo and light-energy expenditure may be common within physical education or coaching during isolated task practices [69,71]. To achieve a *vigorous PA intensity* threshold, 10 attempts per minute at a maximal effort for FMS-related practices were deemed to be needed [69], whereas 20 attempts of short-range kicking were required to reach moderate intensity [71]. Thus, depending upon the effort required and rate of attempts, the practice of individual FMS may produce light, moderate, or vigorous levels of PA.

In contrast to the aforementioned studies on energy expenditure when performing individual FMS [69,71], FMS practice sessions will often include activities in which multiple FMS are performed within the context of a game [72]. According to the compendium of physical activities for children and adolescents, various forms of game play commonly result in levels of vigorous intensity being attained [73]. Several examples of organised games (i.e., basketball, soccer, tennis), in addition to less formal playground and active locomotor play (i.e., hopscotch, freeze tag, sharks and minnows) typically allow children and adolescents to meet the vigorous-intensity threshold across 4 identified age ranges (6 to 9 years old, 10 to 12 years old, 13 to 15 years old, and 16 to 18 years old) [73]. Thus, provided that games are appropriately designed (i.e., number of participants, size of playing area, etc.), it appears that such activities are a viable means of meeting the PA objective of FMS sessions [59].

In relation to skill development, extensive research has shown that practicing FMS in isolation, coupled with appropriate instruction and feedback, can develop children’s skill levels [33,40,74,75]. While the overall time spent in activities is typically reported in these studies, the number of skill executions within that time period is not. Interventions with a larger emphasis on the use of games to enhance FMS have also proved successful [59,71,76,77]. As with interventions based upon practicing skills in isolation, specific information on the rate of skill executions within these games is typically not provided either. Additional information on the rate of skill executions per unit time would prove valuable for practitioners in their design of practice sessions, and potentially to researchers seeking to understand the mechanisms underpinning effective interventions. 

Skill development is an individual process that demands a tailored approach due to varying environmental constraints and the interactions between the task and the individual themselves [5]. Game-based approaches, such as Teaching Games for Understanding [78,79], propose that the teacher or coach draw upon both game forms and more isolated activities as required to meet their individual learners’ needs. Practitioners need to understand how their choice of activity (isolated task or game form; maximum or submaximal effort; etc.) will influence both PA levels and skill development. As such, a singular definitive recommendation regarding which intensity to utilise in a physically active setting to develop FMS is not advisable from the current evidence presented. Instead, Physical Education teachers, sport pedagogues, coaches, and researchers should acknowledge and justify their selections of individual activities and combinations of activities within a session/lesson plan so as to meet learners’ PA and skill development needs [68,69].

### 3.3. Time

When considering recommended FMS guidelines for children and adolescents, the variable of “time” under the FITT acronym can refer to either (1) the duration of time devoted to motor skill instruction and practice across a complete intervention [22,75] or (2) the duration in minutes of motor skill instruction and practice in a singular FMS session of intervention [13].

A previous meta-analysis by Logan et al. (2012), which specifically examined motor skill interventions in children, found that no significant relationship existed between the duration in minutes of the FMS intervention dose and the subsequent effect size of participant FMS improvements post intervention (the intervention–dose response). Many interventions for FMS identified within this meta-analysis were noted as lasting from between 6 and 15 weeks in length, and ranged from 480 to 1440 min (8 to 24 h) total in duration [22]. More recently, Robinson et al. (2017) suggested that 600 min (10 h) of high-quality instruction for pre-schoolers could significantly improve children’s motor competence, and the authors reported similar improvements for children’s FMS performances, regardless of whether participants had received a 660 min (*n* = 27, 13 males, 14 females, mean age = 4.4 years, SD = 0.6 years), 720 min (*n* = 23, 11 males, 12 females, mean age = 4.4 years, SD = 0.4 years), or 900 min (*n* = 25, 13 males, 12 females, mean age = 4.5 years, SD = 0.5 years) dose of FMS instruction as part of the Children’s Health Activity Motor Program (CHAMP) intervention across a 12-week period [75]. In other studies of younger children aged between 2 and 6 years old, evidence would suggest that interventions with a shorter duration (ranging from 1 month to 5 months) have demonstrated significantly higher effect sizes for FMS proficiency when compared with studies of longer durations (6 months or longer) [13]. It has been theorised that the activities provided in the intervention may, over time, become repetitive and monotonous to the children, leading participants to disengage from the intervention and its associated activities [22].

Tompsett et al. (2017) in their updated systematic review of pedagogical approaches used in FMS interventions reported that individual FMS session durations vary widely across the FMS-related literature for children and adolescents, with session durations of: 20 min, 30 min, 40 min, 45 min, 60 min, and 90+ min being reported. Findings from this systematic review observed that both the session duration and the number of sessions per week (frequency) were not associated with FMS proficiency outcomes in participants aged 5–18 years [80]. Many evidence-informed FMS studies with children and adolescents in sport and PE settings have cited that the implementation of two sessions a week appears to promote positive changes in FMS competence, with total session time across the week equalling approximately 60 min [40,74,81,82].

The use, however, of one session a week in the 30–60 min range has also been found to be effective in promoting enhanced skill growth and motor development among children and adolescents [83,84,85]. Aligned to the FITT principle, no clear guidelines for the suggested FMS session(s) time exists in child and adolescent FMS-related research. Existing evidence from above, however, suggests that somewhere between 30 and 60 min per session would appear appropriate for FMS-related skill development. Notwithstanding the relationship between the individual, the task, and the environment in which the motor skill task is performed [86,87,88], the time available for FMS devotion will likely depend on the intervention setting, be that a community-based sports club, or in a school environment through Physical Education classes, for example.

Overall, while no specific FMS guidelines for time have been consistently set within the literature for increased FMS competence in children and adolescents, some impactful research has reported that successful FMS-related interventions appear to comprise at least 600 min of quality instruction time, with effective FMS session durations lasting somewhere between 30 and 60 min per week and being no longer than 6 months overall in duration (particularly when training those in early childhood). Future research regarding the FMS training of children and adolescents is needed, by specifically examining dose–response relationships for meaningful FMS-related intervention guidelines [13,75,80].

### 3.4. Type

FMS development is influenced not just by the frequency, intensity, and time engaged in practice; practitioners must also select the type of practice. In a recent systematic review of the pedagogical approaches used in FMS interventions for children and adolescents [80], it was revealed that FMS interventions are indeed effective at improving FMS proficiency (27 of 29 included studies). Central to the success of these interventions are the deliberate decisions that trained and/or experienced practitioners make when designing and delivering developmentally appropriate activities [2]. This culminating section on the FITT principle’s relationship with FMS will focus on three decisions that practitioners may need to make in relation to type of practice: (i) the nature of guidance, (ii) the level of autonomy afforded to learners, and (iii) the extent to which the FMS are performed in isolation or in the context of a game form (Table 2).

#### 3.4.1. Nature of Guidance

High-quality instruction, practice, and feedback are essential factors for the development of FMS proficiency in children and adolescents [2]. While unstructured, minimally supervised “free play” interventions do appear to lead to improvements in FMS (e.g., [46,89,90]), these improvements are less than those observed in peer groups that receive additional guidance. While some form of additional guidance can enhance learning, this quality instruction may be delivered in different ways [2,91]. For example, direct instruction is where a movement solution is prescribed for the learner by the practitioner. This prescription may be provided in the form of demonstrations, cue words, and/or targeted feedback, all of which is designed to help a child modify their action towards a more proficient pattern (e.g., [74,75]). In contrast, indirect instruction refers to manipulations of the task, equipment, or playing space to elicit behavioural responses from the learner (e.g., [92,93]). For example, instructions to throw “as far as you can” or the use of distant targets may be used to encourage a stepping action and additional trunk rotation within the overarm throw. Importantly, effective indirect instruction does not force a learner towards a single, specific solution, but rather encourages the exploration of alternative movement solutions [94].

One proposed advantage of indirect instruction is that it encourages a learner to become sensitive to the demands of any movement situation, and to adjust their movement accordingly [72]. However, limited research has directly compared direct and indirect instruction while controlling for other variables [93,95], and this research has produced equivocal findings in relation to movement competence, with direct instruction enhancing the development of certain movement components, and indirect instruction enhancing the development of others. The impact of these differing instructional approaches on broader benefits (e.g., intrinsic motivation, creativity) have not been investigated [96]. In addition, many FMS interventions (e.g., SKIP–[97]) utilise both direct and indirect instruction in combination. Effective teachers and coaches can and do use both direct and indirect instruction, often within the same session [91], with the decision depending upon the aim of the activity, and the specific characteristics of the learner and teacher.

#### 3.4.2. Learner Autonomy

The ideal FMS session is one which supports children to become proficient movers while also enhancing their motivation to partake in PA [98]. According to self-determination theory [99,100], an autonomy-supporting learning environment enhances motivation. Within the context of FMS development, autonomy refers to viewing learners as individuals who are deserving of understanding and, within appropriate limits, of choosing the direction of their development [101]. In practical terms, an autonomy-supporting environment is one in which learners are provided with a rationale for activities, their feelings are taken into consideration, and they are provided with as much choice and opportunities for independent action as appropriate in the context [101]. While the provision of a rationale for activities and consideration of learner’s feelings should be present within all FMS sessions, the instructor should determine the appropriate degree of choice to be provided to learners.

Within low-autonomy FMS sessions, the teacher/coach selects the content, duration, and order of activities to be practiced [74,102]. In contrast, during high-autonomy FMS sessions, the learner has a degree of choice about which activities to engage in, which variations of each skill to engage with (e.g., which target to throw at, which object to throw with), how long to spend on each task, and whether they would like feedback on any particular effort [103,104].

Multiple studies have demonstrated the benefits of incorporating learner autonomy within an FMS intervention (e.g., [104,105]). However, in many studies on learner autonomy, the interventions differ on both the level of autonomy provided and on the nature and/or quantity of the instruction provided. Valentini and Goodway (2004b) found benefits for a high-autonomy group relative to a low-autonomy group in terms of heightened variable practice conditions, however the group also differed in the use of private rather than public feedback [105]. The most focused test of autonomy was provided by Robinson and Goodway (2009) [40], who provided highly individualised feedback to participants in both a low-autonomy (the teacher made all decisions about what to practice and when based on their professional judgement) and a high-autonomy group (the learner made all decisions); the groups did not differ in relation to improvements in FMS levels. Taken as a whole, these studies suggest that incorporating learner autonomy is beneficial for FMS development (or at least, does not reduce learning), and may have additional motivational benefits. However, the level of autonomy will vary depending upon the aim of the activity, as well as on learner and teacher characteristics [91]. For example, where an instructor has developed children’s ability to self-direct their play appropriately, higher levels of autonomy can be provided. Furthermore, within a single session, different levels of autonomy may be deemed appropriate for different activities; for example, low autonomy might be appropriate when the priority is to assess children’s performances on a novel activity.

#### 3.4.3. Skill Context

Another decision for practitioners in relation to type of practice relates to the extent to which skills are practiced in the context of games or in isolation. Practicing individual FMS in a station-based structure [33,75] provides children and adolescents with the opportunity to perform numerous practice attempts across a wide range of FMS. Such an approach can prove both engaging and enjoyable as long as a suitable range of activities and variations are provided [85]. In contrast, contextualised skill practices see learners perform multiple FMS in a game context [72,77] applied to achieve a higher-order objective. Such games can be simplified or have elements exaggerated in order to provide an appropriate challenge for learners.

There is a concern that isolated technical exercises may show limited opportunities for transfer to game forms, especially from an ecological dynamics theoretical perspective, where the movement a child demonstrates arises from the specific constraints of the situation [86,106]. In addition, practice in the context of game forms is thought to provide young learners with greater opportunities to demonstrate creativity, problem solving and, decision making [72]. However, for many skills, there are common principles of effective and safe movement which may be best appreciated initially in isolation. Furthermore, the flow of information which guides movement is not just in the external world (e.g., location of target for a throw, intervening obstacles) but also internal to the body in the form of kinaesthetic information from muscles and joints (e.g., absence of knee valgus when landing). Exploring movements in isolation, alongside the implementation of established elements of game-based approaches [107], may facilitate the learner to tune into this kinaesthetic information flow.

Research comparing technical exercises against games skills have reported mixed results. For example, Jarani et al. (2016) reported that 8-year-old Albanian children showed superior improvements in a range of motor skills tests if they performed exercises as individuals (e.g., gait exercises to improve running speed) rather than as small groups (e.g., tag games to improve running speed) [108]. In contrast, Miller et al. (2015) reported that 10-year-old Australian children showed significant improvements in throwing and catching following a games-based intervention compared with lessons featuring a higher proportion of isolated technical training [59]. Thus, as with learner autonomy, it appears that the question facing instructors is not whether isolated or contextualised activities are most effective but rather how and when each type of practice should be applied in order to maximise learning. Indeed, many interventions incorporate both isolated and contextualised activities [24,109,110]. An implication for researchers is to report the degree to which isolated and contextualised skill practices are present within their sessions (e.g., [59]).

This section reviewed three key dimensions of the type of practice and instruction: the nature of guidance, learner autonomy, and the skill context. Each dimension represents a spectrum of activity and instructional design that a teacher or coach can select from depending upon their aims and the needs of the learners. For researchers, additional clarity and consistency is required in the reporting of each dimension of practice type.

## 4. Conclusions

In exploring recommendations for child and adolescent FMS development, an outline of the range of guidelines identified in this narrative review are summarised in Table 3 below using the FITT principle. As a means of equipping practitioners with evidence-based recommendations for child and adolescent FMS development, the use of the FITT principle could be a promising, “user-friendly” strategic approach. As explored in this narrative review, however, a lack of sufficient consistency across published FMS interventions appears to exist across the different studies in terms of intervention frequency, intensity, time, and type. As such, regarding the FITT principle, the evidence is insufficient to provide robust recommendations for practitioners. For these reasons, the guideline ranges presented in Table 3 are therefore not intended to represent robust recommendations but rather to provide a summary of the different findings reported within the available evidence.

It is recommended, rather, that the FITT principle may be used to structure future investigations of child and adolescent FMS interventions, whereby future researchers might report their FMS intervention study designs, in accordance with the FITT principle, to facilitate commonality and comparisons between studies. Such improved reporting and clarity between FMS intervention study designs would strongly contribute to the quality of studies seeking to evaluate the impact of the FITT principle. The authors of the current study, however, strongly suggest that some clear elements need to be considered if seeking to promote quality FMS research in children and adolescents when using the FITT principle.

Reporting the frequency (i.e., dosage) of FMS sessions is a clear necessity for future research. Many FMS intervention studies are evaluated within physical education settings. The duration of such classes and the number of taught classes per week typically vary across countries and continents. Outlining a consistent approach for the frequency of FMS-related physical education lessons may be necessary to examine how the frequency variable could be operationalised in diverse education (or community sport settings). Regarding intensity, some FMS-related research has assessed this variable using portable gas analysers, which evaluate oxygen consumption on a breath-by-breath basis both prior to and during exercise. For FMS practitioners in the field, given that cost is often a prohibitive factor within measurement studies, the use of heart rate monitors or smart watch devices may be considered reasonable alternative measurement devices to gauge exercise intensity during FMS sessions. Future research studies should clearly outline time recommendations when reporting on the FMS-related prescription of intervention studies, with the findings of the current narrative review suggesting that a range of between 30 and 60 min of FMS-specific work per week might be appropriate for the motor development of children and adolescents. Within a school or community sport setting, practitioners are encouraged to target this 30 to 60 min time threshold through allocated classroom or sport-related session times. Clearer specifications on the type of activities used to improve FMS in children and adolescents should be clarified within future research studies as a strategy to identify replicable trends that can be adapted for use within and across countries. It is very important to note that in research and practical settings, the type of instructional offering for promoting individual autonomy within FMS may vary. Such instructional climates may be dependent on the context of the skill, the mode of delivery, and whether additional elements, such as decision making, motivation, etc., may also need to be targeted.

It is recommended by this authorship team that future prospective studies seeking to evaluate the “FITT” principle within FMS environments should provide a clear outline of the frequency (dosage) and time (duration) of the sessions undertaken and specify the instructional methods implemented, with a supportive rationale on the types of activities offered, in addition to measuring intensity. Such consistency in the reporting of FMS interventions for children and adolescents may allow for the future provision of evidence-informed, FMS-related recommendations for use by practitioners, including Physical Education teachers, sport pedagogues, coaches, parents, guardians, and researchers.

## Figures and Tables

**Table 2 ijerph-20-03278-t002:** Dimensions of practice type to enhance fundamental movement skills.

Dimension			
**Nature of Guidance**	**Direct instruction**: movement solution specified through some combination of demonstration, instruction, physical guidance, and/or prescriptive feedback.	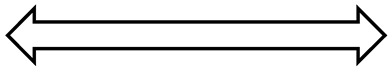	**Indirect instruction**: manipulation of constraints (e.g., distance from target; object to be thrown) to encourage alternative behaviour/exploration.
**Learner Autonomy**	**Teacher selects** content, sequence, and duration of practice activities.	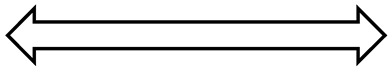	**Learner selects** content, sequence, and duration of practice activities.
**Skill Context**	**Isolated** technical practice (exercise), often with task decomposition.	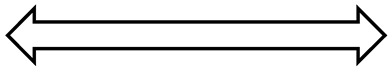	**Contextualised** skill practice (game), often with simplification and/or exaggeration.

**Table 3 ijerph-20-03278-t003:** The FITT formula: identified guideline ranges for the FMS development of children and adolescents.

Fundamental Movement Skills	
**F**	**Frequency:** At a minimum 2 times per week (unknown if higher dosages of FMS-related frequencies per week bring about additional motor competence and/or motor skill development).
**I**	**Intensity:** Moderate-to-vigorous thresholds, with a priority towards object-control skills. Desired FMS intensities can be reached through direct and indirect instructional practice pedagogies.
**T**	**Time:** Aim for between 30 to 60 minutes of FMS-related activities per week, striving for at least 600 minutes of total intervention or overall program dosage time.
**T**	**Type:** Avail of FMS teacher/coach expertise, supported by parents/guardians. Practice FMS regularly in structured (games, stations) and unstructured activities (free play).

## Data Availability

Not applicable.

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
