# Peer review of "Exploring Recommendations for Child and Adolescent Fundamental Movement Skills Development: A Narrative Review"

_ijerph, 2023, doi:10.3390/ijerph20043278_

Round 1
Reviewer 1 Report
The subject of this manuscript is current and is an important concern for readers.
I will leave some comments and suggestions that may be important to make the manuscript more complete and clear to our readers.
Introduction:
It is very well constructed, being clear, precise and concise.
It should only add more information and bibliography regarding the importance of fundamental motor skills in motor development.
Add the expected results and major limitation at the end of the introduction.
Conclusion:
You should explain the statement better
"As explored in this narrative review, however, a lack of sufficient consistency in the FMS intervention reporting appears to exist across studies in terms of intervention frequency, intensity, time and type." and "As such, with regards to the FITT principle, the evidence is insufficient to provide robust recommendations for practitioners"
It should clearly identify again the objective of the study and what the major conclusion was obtained from this study, referring to the limitations of this investigation and what are the suggestions for future studies within the same theme.
Author Response
"Please see the attachment."

Reviewer 2 Report
Thank you for the opportunity to review this paper on recommendations for child and adolescent FMS interventions.
This is a well-written, interesting manuscript on a topic of value. I am happy to recommend this for publication, though I do see a few minor points worth working on first. I provide my comments according to the different sections below.
Introduction
The introduction is well-written and makes a clear case.
My one concern is that it is not clear to the reader how you searched for and identified the literature. I understand the narrative and practical orientation of the paper, and I don't require a full, PRISMA-type methodology section. However, your search strategy (i.e. databases, search strings) and inclusion criteria (e.g. thematic focus, date, language) should be presented.
Discussion/Conclusion
Though I understand that a narrative review is already discursive in nature, I am left wanting more from your conclusion - the implications and future directions are rather brief. In such a practically-oriented paper, I find it is of extra importance to present a clear bottom line and implications. In particular, I would recommend two things.
First, you could still summarise your findings against the FITT model in a table or figure (even if you only present ranges as opposed to clear recommendations). As a reader, I felt there was a lot of valuable information, and a condensed, more visual summary would be helpful.
Second, you talk about using the FITT principle to structure studies. I think that you should go into more depth here and provide concrete thoughts or examples of how that would look like in practice.
Author Response
"Please see the attachment."

Round 2
Reviewer 1 Report
The authors carried out all the suggestions proposed by me to improve the manuscript and thus be more perceptible to the readers of our journal.
Therefore, I understand that the conditions for accepting the respective publication are met.
Best Regards,